# The Role of DJ-1 in Cellular Metabolism and Pathophysiological Implications for Parkinson’s Disease

**DOI:** 10.3390/cells10020347

**Published:** 2021-02-07

**Authors:** Pauline Mencke, Ibrahim Boussaad, Chiara D. Romano, Toshimori Kitami, Carole L. Linster, Rejko Krüger

**Affiliations:** 1Translational Neuroscience, Luxembourg Centre for Systems Biomedicine, University of Luxembourg, 4365 Esch-sur-Alzette, Luxembourg; ibrahim.boussaad@uni.lu; 2Biospecimen Research Group, Integrated Biobank of Luxembourg, Luxembourg Institute of Health (LIH), 3531 Dudelange, Luxembourg; chiara.romano@ext.uni.lu; 3Enzymology & Metabolism, Luxembourg Centre for Systems Biomedicine, University of Luxembourg, 4365 Esch-sur-Alzette, Luxembourg; carole.linster@uni.lu; 4RIKEN Outpost Laboratory, Luxembourg Centre for Systems Biomedicine, University of Luxembourg, 4365 Esch-sur-Alzette, Luxembourg; toshimori.kitami@uni.lu; 5Parkinson Research Clinic, Centre Hospitalier de Luxembourg (CHL), 1210 Luxembourg (Belair), Luxembourg; 6Transversal Translational Medicine, Luxembourg Institute of Health (LIH), 1445 Strassen, Luxembourg

**Keywords:** DJ-1, Parkinson´s disease, metabolism

## Abstract

DJ-1 is a multifunctional protein associated with pathomechanisms implicated in different chronic diseases including neurodegeneration, cancer and diabetes. Several of the physiological functions of DJ-1 are not yet fully understood; however, in the last years, there has been increasing evidence for a potential role of DJ-1 in the regulation of cellular metabolism. Here, we summarize the current knowledge on specific functions of DJ-1 relevant to cellular metabolism and their role in modulating metabolic pathways. Further, we illustrate pathophysiological implications of the metabolic effects of DJ-1 in the context of neurodegeneration in Parkinson´s disease.

## 1. Introduction

Despite having been originally identified as an oncogene upregulated in different types of cancer, DJ-1 has also been clearly assigned as a causative factor for neurodegeneration in rare inherited forms of Parkinson’s disease (PD) [1,2]. DJ-1 is encoded by the *Park7* gene and is ubiquitously expressed. In the human brain, the DJ-1 protein is abundantly expressed in reactive astrocytes and to a lower extent in neurons [3,4]. In the mouse CNS, DJ-1 transcript and protein were shown to be expressed at similar levels in neurons, astrocytes, microglia and oligodendrocytes [5].

DJ-1 is a small protein of 189 amino acids that forms homodimers. DJ-1 belongs to a functionally diverse protein superfamily whose members are all characterized by an α/ β-flavodoxin fold [6]. Human DJ-1 displays the characteristic central β-sheet surrounded by eight α-helices (sandwiched structure) and the highly conserved cysteine (Cys) 106 residue in the ‘‘nucleophile elbow’’ pocket [7]. The thiolate group of Cys106 can be oxidized to sulfinate (-SO2−) and sulfonate (-SO3−) under oxidative stress conditions [8]. This oxidation shifts the isoelectric point of the protein, promoting intracellular relocation of the enzyme to the mitochondria as well as its cytoprotective function [9]. DJ-1 was shown to be involved in many processes, including regulation of apoptosis and pro-survival signaling, autophagy, inflammatory responses and protection against oxidative stress [10,11]. DJ-1 was also shown to have chaperone activity [12] and to act as a glyoxalase III, able to detoxify reactive dicarbonyl species such as glyoxal (GO) and methylglyoxal (MGO) in a glutathione-independent way [13] (Figure 1). Recently, an additional debate emerged about a potential deglycase activity of DJ-1 [14].

There are many studies pointing towards the effects of DJ-1 on cellular metabolism at different levels, but the underlying molecular mechanisms remain for the most part poorly understood.

Metabolic alterations are often linked to pathogenic conditions, and first links have been established for PD [15,16]. Indeed, impaired energy metabolism is associated with PD, as the inactivation of the electron transport chain complex 1 is classified as one of the hallmarks of PD [17], and the importance of metabolic dysfunction in PD has been increasingly discussed [15,16]. To what extent the loss of DJ-1 function in PD could contribute to disease pathogenesis by impairing or dysregulating energy or other parts of metabolism remains elusive.

In this review, we will summarize the current knowledge on the modulation of cellular metabolism by DJ-1 and the potential pathophysiological implications for neurodegeneration in PD. For other disease conditions associated with DJ-1 or for disease connections to broader functions of DJ-1, we refer readers to excellent reviews focusing on cancer [18], diabetes [19,20], inflammatory diseases [21] and Parkinson’s disease [22].

## 2. DJ-1 in Parkinson’s Disease

Parkinson’s disease (PD) is the second most common neurodegenerative disease that affects 1–2% of the population over age 60 world-wide with increasing prevalence [23,24]. Cardinal symptoms of PD include tremor, bradykinesia, rigor and postural instability resulting from loss of dopaminergic neurons in the substantia nigra pars compacta [25]. Cellular hallmarks of PD include intraneuronal proteinaceous inclusions, Lewy bodies and neurites that contain α-synuclein as a major component [26]. Familial cases of PD account for up to 20% of all PD cases, of which approximately 5–10% have known monogenetic causes. To date, mutations in *SNCA*, *Parkin*, *PINK1, DJ-1*, *LRRK2* and *ATP13A2* are known to be causative for familial PD, but each of these monogenetic forms of PD is rather rare [24].

In 2003, Bonifati and colleagues found a large (about 14 kb) deletion and a missense mutation (Leucine166Proline, L166P) in the *Park7* gene in a Dutch and Italian family, respectively, which led to the identification of *Park7* as a causative gene for familial PD with recessive inheritance [2]. Since then, more than 20 DJ-1 mutations have been associated with early-onset PD. The impact of pathogenic single amino acid substitutions on the enzyme structure has been analyzed quite systematically [6]. The L166P mutation, for instance, affects the DJ-1 protein structure and function [27]. The lysine 166 residue is located in the center of the α-helix 7, which is located near the dimeric interface [8,28,29]. Introduction of a proline residue into a helix structure is not well tolerated as it increases the molecular rigidity, leading to a break in the helix and, predictably, C-terminal unfolding of the DJ-1 L166P variant [30]. The latter loses the dimerization property observed in the wildtype (WT) protein, forming unstable monomers that are prone to degradation [6]. The L10P and P158 mutations also disrupt the homodimerization of DJ-1 [31]. Other missense mutations, such as M26I and A104T, have less severe impact on the protein structure and do not prevent homodimerization [6]. However, both mutant proteins have been reported to be unstable, especially in the cellular environment [6,32]. Very recently, a DJ-1 mutation initially predicted to lead to a missense mutation (E64D) was shown to lead to decreased intracellular protein levels due to U1-dependent pre-mRNA mis-splicing [33]. Taken together, the information available on PD-associated DJ-1 variants so far seems to converge on the notion that a resulting loss-of-protein function explains their pathogenicity [34].

Patients with DJ-1 mutations develop early onset, slowly progressive parkinsonism, and most of them present with typical PD [35]. However, atypical forms with clinical symptoms related to other neurodegenerative diseases, i.e., amyotrophic lateral sclerosis and dementia, were described [36]. Cellular phenotypes of DJ-1 loss of function are predominantly mitochondrial dysfunction [37] and a reduced dopaminergic differentiation potential of PD patient-derived DJ-1-deficient cells [33]. These initial studies suggested that metabolic functions of DJ-1 may be important for the PD pathogenesis.

The mitochondrial phenotype in DJ-1-deficient models as well as other metabolic alterations induced by pathogenic DJ-1 mutations (summarized in Table 1) support the notion that PD pathogenesis involves an important metabolic component.

### 2.1. DJ-1 in Other Diseases

Despite its clear pathogenic role in PD, DJ-1 is also involved in diseases such as cancer [50], obesity, insulin resistance and type 2 diabetes mellitus (T2DM) [19,51,52].

In cancer, DJ-1 was found to play an important role in tumor progression of various cancer types through regulation of prosurvival signaling, for example via negative regulation of the tumor suppressor p53 [53]. DJ-1 regulates cell survival and proliferation via the extracellular signal-regulated kinase (ERK1/2) pathway and the phosphatidylinositol-3-kinase (PI3K)/Akt pathway. It counteracts apoptosis by inhibiting the activation of apoptosis signal-regulating kinase 1 (ASK1) and of the mitogen-activated protein kinase kinase kinase 1 (MEKK1/ MAP3K1) apoptotic cascades [10]. For example, in glioblastoma multiforme (GBM), DJ-1 levels were shown to be increased [54]. Hinkle and colleagues found also that immunostaining intensity of DJ-1 in GBM tissue varied directly with strong nuclear p53 expression and inversely with EGFR amplification [54]. Since DJ-1 negatively regulates pro-apoptotic p53 and EGFR signaling, these findings suggest that DJ-1 might be involved in promoting gliomagenesis.

The role of DJ-1 in glioblastoma is especially interesting as glioblastoma is thought to originate from astrocytes, which are also involved in PD pathogenesis, and increased DJ-1 expression is pathogenic for glioblastoma, but protective in models of PD [18].

In the following, we will focus on discussing the functions of DJ-1 in the modulation of cellular metabolism in the context of neurodegeneration in PD.

### 2.2. DJ-1 and Its Chaperone Function

Chaperone activity is essential to promote correct protein folding. Under “critical” situations, such as during oxidative stress, it becomes important to counteract protein denaturation and aggregation caused by oxidative damage. DJ-1 belongs to the DJ-1/ThiJ/PfpI superfamily and, albeit belonging to a different phylogenetic clade, shows structural similarities with another member of that superfamily, the heat shock protein Hsp31 [6,9]. The latter is a microbial protein and has been studied more extensively in the model organism *Saccharomyces cerevisiae* (i.e., budding yeast). Hsp31 is involved in the protection against reactive oxygen species (ROS), as confirmed by the *hsp31*Δ yeast strain, which is more sensitive to linoleic acid hydroperoxide and other ROS generating agents [55]. In addition to its ROS scavenging function, Hsp31 also acts as a chaperone for an array of proteins, including α-synuclein [56]. This small protein, encoded by the *SNCA* gene, exists under a native unfolded form in the cytoplasm and a more organized α-helical conformation when associated with cellular membranes [57]. Pathological conformational changes in α-synuclein lead to the formation of protein fibrils and Lewy bodies, the characteristic intraneuronal pathological inclusions in brains of PD patients [58]. The Hsp31 protein has been shown to prevent aggregation of α-synuclein in vitro and in living yeast cells expressing toxic levels of human α-synuclein [56,59]. This chaperone activity of Hsp31 was not dependent on the protein’s glyoxalase activity [56].

A similar, although weaker chaperone activity against α-synuclein aggregation (in vitro, in the yeast α-synuclein model, and in murine neuroblastoma cells) has been reported for human DJ-1, and PD-causing DJ-1 mutations were shown to decrease the interaction with α-synuclein [12,56,59]. Burbulla and colleagues found that intracellular levels of soluble and insoluble α-synuclein were elevated in iPSC-derived human neurons from homozygous DJ-1 mutation carriers [60]. Kumar and colleagues reported that partially oxidized DJ-1 exposes an adhesive surface, which can sequester monomers of α-synuclein and block early stages of α-synuclein aggregation and also restrict the elongation of α-synuclein fibrils [61]. Importantly, and in line with this chaperoning function of DJ-1, patients with an autosomal recessively inherited form of juvenile PD due to homozygous loss-of-function mutations in the DJ-1 gene indeed show Lewy bodies in affected brain regions post-mortem [62].

Interestingly, Solti and colleagues found that DJ-1 itself can aggregate into β-sheet structured soluble and fibrillar aggregates in vitro under physiological conditions and accelerated when oxidized at its Cys106 residue [63]. They observed that as a result of the aggregation of DJ-1, its glyoxalase function was abolished [63]. In addition, DJ-1 aggregates were localized within Lewy bodies, neurofibrillary tangles and amyloid plaques in post-mortem brain tissue from PD and Alzheimer’s patients [63]. The authors discuss that PD-associated loss of DJ-1 function in sporadic PD could be caused by its aggregation [63].

### 2.3. DJ-1 and Its Enzymatic Function

The reactive dicarbonyls glyoxal (GO) and methylglyoxal (MGO), which are formed in cells from various sources, including lipid peroxidation and the glycolytic triose-phosphate intermediates, respectively, can damage biomolecules via glycation. The latter are spontaneous chemical reactions between amino or thiol groups of, e.g., proteins or nucleotides and the carbonyl carbon of aldehyde and ketone groups in sugars and sugar derivatives [64]. The resulting adducts can react further to form “advanced glycation end products” (or AGEs), which accumulate over time and are considered as an inevitable component of the aging process [65]. An accumulation of AGEs can interfere with biological function and result in cellular damage [65]. Dicarbonyl damage has been associated with several diseases, including T2DM and PD [66]. The glutathione-dependent glyoxalase system is the major cellular protection mechanism against dicarbonyl damage. It converts GO and MGO to glycolic and lactic acid, respectively, through the consecutive action of glyoxalase I and glyoxalase II in the presence of catalytic amounts of reduced glutathione (GSH) [67]. An additional glutathione-independent glyoxalase activity, named glyoxalase III, has first been detected in *Escherichia coli* [68] and was subsequently identified as Hsp31 [69], a member of the DJ-1 superfamily (already mentioned above because of its chaperone activity). Robust glyoxalase III activity has since also been detected in the yeast species *S. cerevisiae*, *Schizosaccharomyces pombe* and *Candida albicans* [70,71,72]. A weaker glyoxalase III activity has also been measured for human DJ-1 and its *Caenorhabditis elegans* homologs [13,72] (Figure 1). DJ-1 conferred protection against toxic effects of glyoxal treatment in mouse embryonic fibroblasts, SH-SY5Y cells and *C. elegans* worms [13]. Given the relatively weak glyoxalase activity of DJ-1, compared to the highly active and ubiquitous glyoxalase I/II system, the physiological relevance of DJ-1 for (methyl)glyoxal detoxification remains, however, questionable.

In addition to its glyoxalase function, Richarme and colleagues proposed that DJ-1 could act as a novel deglycase that repairs methylglyoxal- and glyoxal-glycated amino acids, proteins, nucleotides and nucleic acids by acting on early glycation intermediates and releasing lactate or glycolate [38,39]. Matsuda and colleagues suggested that DJ-1 protects glutathione and coenzyme A (CoA) from aldehyde attack [40]. They found that glutathione (GSH), CoA and β-alanine (a CoA precursor) are recovered from methylglyoxal-adducts by recombinant human DJ-1 purified from *E. coli*. During this process, MGO was converted to L-lactate rather than the D-lactate produced by the conventional glyoxalase I/II system. PD-associated DJ-1 mutations (L10P, M26I, A104T, D149A and L166P) were shown to impair or abolish this detoxification activity, suggesting that further dissection of the methylglyoxal-adduct hydrolase activity of DJ-1, which protects low-molecular thiols from dicarbonyl damage, may be a promising research direction to progress in our understanding of PD pathophysiology [40].

Jun and Kool recently published a comprehensive review that explains the controversial debate around the deglycase function of DJ-1, concluding that further studies are needed to clarify this potential function of DJ-1 [14]. As a central question, it remains to be determined whether DJ-1 has a direct deglycase activity or whether the observed deglycation results from removal of the small aldehydes (via the glyoxalase activity described above), which are in rapid equilibrium with the glycated adducts.

### 2.4. DJ-1 and Mitochondrial Function

In addition to its enzymatic functions, DJ-1 plays an important role in mitochondrial homeostasis. Mitochondria are the essential organelles for energy metabolism, as they provide the cell with ATP via the tricarboxylic acid (TCA) cycle and subsequent oxidative phosphorylation (OXPHOS) through the electron transport chain. Thus, changes in mitochondrial homeostasis can have drastic effects on the energy metabolism of the cell, especially on neurons that have a high energy consumption for maintaining synaptic activity.

It is well documented that changes in mitochondrial organellar homeostasis, as indicated for example by altered mitochondrial morphology, are associated with different in vitro and in vivo models of PD [46]. The *Drosophila melanogaster* genome encodes two DJ-1 homologs: DJ-1α and DJ-1β [47]. In *Drosophila,* the impact of loss of DJ-1 on mitochondrial quality control may involve two other important effectors well known in the context of PD: PINK1 and Parkin. It was shown that PINK1 and Parkin are both implicated in a common pathway that regulates mitochondrial dynamics and cell survival [73,74,75]. Yang and colleagues found that downregulation of PINK1 has deleterious effects in the fly model: flight ability is compromised by the flight muscle degeneration, and dopamine levels in the brain decrease with age. Electron microscopy analysis of tissues revealed swollen mitochondria, also in agreement with low ATP levels [76]. The overexpression of Parkin rescued loss-of-PINK1 related phenotypes, further supporting that Parkin acts downstream of PINK [76]. It is still a debate whether and how DJ-1 may integrate into this pathway, and what relation exists between PINK1, Parkin and DJ-1 in the maintenance of mitochondrial homeostasis. In *Drosophila,* there are controversial data for PINK1 knockout (KO) models concerning a selective rescue of PINK1 mutants by DJ-1 [73,76]. Even if one study places DJ-1 homologs downstream of *Drosophila* PINK1 with an expression-level dependent rescue of loss of PINK1 function, the fact that DJ-1 cannot rescue Parkin mutants and Parkin cannot rescue loss of DJ-1 in flies indicates that DJ-1 does not act within the same pathway. Here, it has been suggested that DJ-1 acts in a pathway parallel to that of PINK1/Parkin [73].

Another modulator of mitochondrial dynamics is dynamin-like protein 1 (DLP1), or Drp1, a regulator of mitochondrial fission. It was shown that the levels of Drp1 were increased in DJ-1 mutant M17 human neuroblastoma cells [37]. The knockdown of Drp1 in DJ-1 mutant cells resulted in a rescue of the abnormal mitochondrial morphology and associated mitochondrial/neuronal dysfunction. Other studies confirmed the fragmentation phenotype, but remained controversial about the impact of modulations of Drp1, with normal total Drp1 [42,43] or decreased Mfn1 levels [41] in different models of reduced DJ-1 function. Therefore, increased fission related to loss of DJ-1 could be caused by an insufficient energy supply to maintain mitochondrial fusion processes [42] or based on an impaired ER-mitochondria communication related to altered tethering of membranes from both organelles [77]. Taken together, these controversial data suggest that DJ-1 might not regulate mitochondrial dynamics primarily via modulation of Drp1 expression, but that PD-associated loss of DJ-1 function may cause impaired mitochondrial function with impact on morphology and clearance of mitochondria based on multiple pathways [37]. PD-associated loss of DJ-1 function was found to be associated with reduced basal autophagy in mice [41,42] and M17 [43] cells, which was corroborated by an accumulation of dysfunctional mitochondria [41,42,43], eventually creating a vicious circle of dysfunctional mitochondria that accumulate and cause further cellular damage. Interestingly, GSH supplementation of DJ-1-deficient cells reversed both mitochondrial and autophagic alterations, which implies that DJ-1 may play an even more important role in mitochondrial function under oxidative stress and that it could influence mitochondrial dynamics and autophagy indirectly [78].

In addition to its role in the regulation of mitochondrial dynamics, DJ-1 was also shown to regulate the association of mitochondria and the endoplasmatic reticulum (ER).

Liu and colleagues found that DJ-1 localized to the mitochondria-associated membrane in vitro and in vivo. More specifically, they observed that DJ-1 physically interacts with the IP3R3-Grp75-VDAC1 complexes at the mitochondria-associated membrane and that DJ-1 is an important component of that complex. In the absence of DJ-1, the complex formation was disrupted and ER-mitochondria association was reduced. This phenotype was rescued by the expression of WT DJ-1, but not by the familial PD-associated L166P mutant [79], suggesting that impaired ER–mitochondria interaction plays a role in DJ-1-associated PD pathogenesis [79]. Overall, DJ-1 can regulate mitochondrial function via changes in mitochondrial clearance as well as through ER–mitochondria interaction.

Given the importance of DJ-1 for the maintenance of mitochondrial function, deficiency of this protein should directly impact cellular metabolism. In addition, DJ-1 acts as a scavenger of ROS, which play an important role as signaling molecules in cellular metabolism, but can also be deleterious when chronically increased.

### 2.5. DJ-1 and ROS Signaling

ROS signaling contributes to physiological homeostasis, but when dysregulated it contributes to disease pathogenesis via alterations in signaling cascades controlling metabolic function.

Cytosolic ROS are produced predominantly by the NADPH oxidase (NOX) family enzymes. ROS produced by NOX enzymes induce the expression of hypoxia inducible factor 1α (HIF1α), which activates the expression of glucose transporter 1 (GLUT1) and the activity of hexokinase, thereby upregulating glycolysis during hypoxia [80]. Mitochondrial ROS are mainly produced by the electron transport chain complexes, and they can also stabilize HIF1α and regulate cell proliferation [81]. DJ-1 deficiency, which is associated with increased ROS levels [45], disturbs these hypoxia response pathways. Parsanejad and colleagues have shown that loss of DJ-1 resulted in decreased HIF1α levels upon hypoxia in primary cortical neurons [82].

DJ-1 can regulate ROS levels via nuclear factor erythroid 2–related factor 2 (Nrf2), a transcription factor that activates genes involved in oxidative stress response as well as in NADPH and ATP production [48]. Clements and colleagues have shown that DJ-1 induces the dissociation of Nrf2 from its inhibitor Keap1 (Kelch-like ECH-associated protein 1), which leads to nuclear translocation of Nrf2 and binding to antioxidant response elements (AREs) in MEF cells [48]. By inducing Nrf2 activation, DJ-1 protects neurons against oxidative stress [48].

However, in another study involving primary cortical neurons, Nrf2 could still be activated in DJ-1 deficient mice, suggesting that DJ-1 is not required for Nrf2 activation at least in this cell type [49].

Structurally, the Cys106 of DJ-1 is preferentially oxidized in cells exposed to oxidative stress [8,83] and is generally known to be the key residue involved in DJ-1 antioxidative function [84]. This is why DJ-1 is also referred to as an “oxidative stress sensor” within cells whose stable Cys106-SO2- modification induces the mitochondrial relocalization of DJ-1. The latter leads to the protection from oxidative stress-induced cellular damage [9,85], one of many mechanisms through which DJ-1 exerts its neuroprotective function [83]. Importantly, the oxidation status of DJ-1 Cys106 seems to be biphasic. Cys106 resides in a pocket, and the transition from oxidized Cys106-SO_2_^–^ to over oxidized Cys106-(e.g., SO_3_^–^) can change the local conformation of the protein leading to destabilized dysfunctional DJ-1 [53]. Therefore, it can be envisaged that the composition of DJ-1 complexes under acute or mild oxidative stress will be different from the one that can be found under conditions that are chronically and excessively oxidizing, thus changing the physiological response of the cell, for example, from pro-survival to apoptotic [53]. Piston and colleagues found that the levels of total DJ-1 and of DJ-1 oxidized at Cys106 were decreased in the cortex of idiopathic PD brains when compared to age-matched control tissue. Moreover, DJ-1 formed high molecular weight complexes in the human brain, which was dependent on the oxidation state of Cys106 [86]. Piston and colleagues also found that proteins involved in RNA transcription/translation seemed to be associated with the complexes of DJ-1, and the composition of the complexes was affected by the oxidation status of DJ-1. Interestingly, these transcripts were associated with the catecholamine system, including dopamine metabolism [87].

### 2.6. DJ-1 and Serine/Glutathione/Glutamine Metabolism

Meiser and colleagues used stable isotope-assisted metabolic profiling to investigate the effect of a functional loss of DJ-1 in LUHMES cells, a human dopaminergic neuronal culture model, and found that DJ-1-deficient neurons exhibit decreased glutamine uptake and reduced serine biosynthesis (Figure 1). Both glutamine and serine are required to generate L-glutamyl-L-cysteine, an important precursor of the antioxidant molecule GSH. Serine is converted into cysteine via the transsulfuration pathway, and glutamine is converted into glutamate, and they together form L-glutamyl-L-cysteine via glutamate cysteine ligase. Downregulation of these pathways, as a result of loss of DJ-1, leads to an impaired antioxidant response [88]. In line with the decreased serine biosynthesis in DJ-1-deficient cells, loss of DJ-1 in MEFs decreased protein and transcript levels of ATF4 [89], a transcription factor that activates serine biosynthesis genes including PSPH, PHGDH and PSAT1 [90] (Figure 1).

Meiser and colleagues also reported an increased sensitivity to H2O2-induced oxidative stress, resulting in a 30% decrease in reduced GSH levels and higher ratio of oxidized (GSSG) to reduced GSH in DJ-1 KO mice [88]. The observed that a decrease in GSH levels and a decrease in enzyme levels of the GSH homeostasis pathways are caused by the loss of DJ-1 and result in insufficient ROS quenching in DJ-1-deficient neurons [88].

Zhou and colleagues found that overexpression of DJ-1 in the N27 rat dopaminergic cell line and in primary dopaminergic neurons protected these cells from death induced by H2O2 and 6-hydroxydopamine [91]. They found that DJ-1 prevents cell death by increasing the level of glutamate cysteine ligase, a rate-limiting enzyme for GSH biosynthesis. The cytoprotective effect of DJ-1 was absent when GSH synthesis was blocked, but the protection could be restored by adding exogenous GSH.

These data indicate that DJ-1 protects dopaminergic neurons from oxidative stress-induced cell death by upregulating GSH synthesis [91] (Figure 1).

In another study, the effect of oxidative stress on GSH metabolism and DJ-1 protein was investigated. Downregulation of glutaredoxin (GRX), but not GSH depletion, resulted in a decrease in DJ-1 protein, translocation of Daxx (a death-associated protein) from the nucleus and subsequent cell death. Daxx translocation and cytotoxicity was prevented by overexpression of DJ-1. Protease inhibitors prevented the decrease in DJ-1 level. Residual DJ-1 was present in a reduced state, which implies that when DJ-1 was oxidized, it was degraded through proteolysis. Thus, the loss of DJ-1 occurring through its oxidative modification and subsequent proteolysis may contribute to PD pathogenesis [92].

In vivo, Lopert and colleagues found that brains from DJ-1 KO mice had an increase in mitochondrial respiration-dependent H_2_O_2_ consumption when compared to control mice [93], indicating that DJ-1 KO mice had a higher capacity to eliminate H_2_O_2_ compared to WT control. However, DJ-1 KO mice showed an increase in oxidized GSSG to reduced GSH ratio and a decrease in mitochondrial glutathione reductase activity, suggesting that other factors may be responsible for increased H_2_O_2_ consumption. The authors instead found an increase in mitochondrial thioredoxin 2 (TRX2) activity and mitochondrial glutaredoxin activity in DJ-1 KO brain compared to WT controls. Therefore, the observed increase in the enzymatic activities of mitochondrial TRX2 and GRX could be causal for the observed increased H_2_O_2_ consumption in mitochondria of brains from DJ-1 KO mice, and this might be an adaptive response to chronic DJ-1 deficiency [93].

### 2.7. DJ-1 and the Regulation of Glycolysis and the TCA Cycle

There is increasing evidence for a direct involvement of DJ-1 in cellular energy metabolism via effects on glycolysis and the TCA cycle. Here, we will describe this involvement of DJ-1 starting from glycolysis, onto TCA cycle and OXPHOS, and finally to signaling and transcriptional regulation of metabolism.

Piston and colleagues analyzed DJ-1 WT high molecular weight complexes from dopaminergic SH-SY5Y cells and identified that glyceraldehyde 3-phosphate dehydrogenase (GAPDH) forms a complex with DJ-1 [86] (Figure 1). GAPDH is a glycolytic enzyme that converts glyceraldehyde 3-phosphate into 1,3-bisphosphoglycerate. Importantly, knockdown of DJ-1 or expression of the PD-associated DJ-1 variant L166P resulted in the absence of high molecular weight DJ-1 complexes [86]. It is not known what the consequences of the interaction of DJ-1 and GAPDH are, but it suggests a possible modulation of the glycolytic pathway by DJ-1 via regulation of GAPDH.

Ozawa and colleagues performed a 2D gel electrophoresis-based proteomic analysis of brain tissue from DJ-1 deficient mice and found a significant change in protein expression of pathways related to energy production including glycolysis, creatine pathway, mitochondrial TCA cycle, and ROS signaling pathway [94]. According to their analysis, spots of proteins such as PDH were decreased in DJ-1 KO compared to WT mice (Figure 1). PDH is a key enzyme in the regulation of metabolism as it connects glycolysis and the TCA cycle and determines whether pyruvate is converted into acetyl-CoA or reduced into lactate. Consistent with a decrease in PDH protein level, the authors found a decrease in mitochondrial ATP production rate in DJ-1 KO SH-SY5Y cells, although a compensatory increase in lactate production was not detected.

DJ-1 was also shown to control PDH activity in CD4 regulatory T cells (Tregs). DJ-1 binds to PDH-E1 beta (PDHB), which leads to the inhibition of the phosphorylation of PDH-E1 alpha (PDHA), thereby promoting PDH activity and OXPHOS [95] (Figure 1).

DJ-1 depletion caused impaired Treg proliferation and cellular maintenance in older mice [95]. DJ-1 was also shown to interact with PDHB in HEK 293, SH-SY5Y and in the mouse brain using immunoprecipitation and mass spectrometry of the mitochondrial protein interactome [96]. However, the effect of this direct protein–protein interaction on the activity of the PDH enzyme still needs to be investigated in neuronal cells.

As mentioned earlier, DJ-1 was found in different subcellular compartments, and it is claimed that the localization of DJ-1 determines its function. Cali and colleagues used HeLa cells to analyze whether DJ-1 metabolic function depends on its localization and activity within the mitochondria. This study revealed that a small DJ-1 fraction is located within the mitochondrial matrix [97] and that it consistently increases upon nutrient depletion. Targeting of DJ-1 to the mitochondrial matrix enhanced mitochondrial and cytosolic ATP levels. Interestingly, overexpression of DJ-1 pathogenic mutants (C106T, M26I and L166P) did not enhance ATP levels, and these mutants were unable to translocate into the mitochondrial matrix upon nutrient depletion, suggesting that DJ-1 localization is also critical for regulating cellular metabolism [98].

It was shown by Chen and colleagues that DJ-1 binds directly to the F1FO ATP synthase β-subunit in HEK293T cells. The interaction of DJ-1 with the β-subunit increased the efficiency of ATP production [99]. Guzman and colleagues found that DJ-1 deficiency in murine neurons resulted in decreased mRNA levels of the uncoupling proteins Ucp5 and Ucp4 and compromised mitochondrial uncoupling in ex vivo brain slices of DJ-1 KO mice [100]. These data provide additional support for an important role of DJ-1 in the modulation of mitochondrial energy production.

Weinert and colleagues describe an interaction between DJ-1 and signaling molecule 14-3-3β that regulates the localization of DJ-1 in a hypoxia-dependent manner, either to the cytosol or to mitochondria [101]. In HEK293T cells, the authors found that DJ-1 is preferentially located in the cytosol by forming a complex with 14-3-3β. Upon cellular stress, including hypoxia or dissipation of mitochondrial membrane potential, DJ-1 dissociates from 14-3-3β and enters mitochondria. In primary neurons, however, DJ-1 was already found to be abundant in mitochondria, suggesting that different cell types have different baseline levels of mitochondrial DJ-1 relative to cytosolic DJ-1 [48]. Furthermore, it was shown that DJ-1 knockdown decreased and 14-3-3β knockdown increased mitochondrial membrane potential in HEK293T cells, suggesting that DJ-1 localization may regulate the energetic potential of mitochondria [101].

Another important molecular interaction concerning metabolic control involves DJ-1 and the Von Hippel Lindau (VHL) protein. VHL ubiquitinates HIF-1α in normoxia, leading to HIF-1α degradation, thus preventing hypoxic response. DJ-1 was shown to bind VHL in SH-SY5Y cells and to suppress VHL ubiquitin ligase activity, thereby blocking VHL-mediated degradation of HIF-1α [82]. Under hypoxia, DJ-1 KO resulted in lower HIF-1α level and showed increased sensitivity to oxidative stress induced by MPP+ in cortical neurons [82]. Increased sensitivity to MPP+ was rescued by HIF-1α overexpression, suggesting that DJ-1 is important for activating the HIF-1α-dependent oxidative stress response.

However, HIF-1α activation is also known to reprogram cellular metabolism by upregulating glycolytic gene expression and by inhibiting pyruvate entry into the TCA cycle, thus mediating a shift from OXPHOS to glycolysis and attenuating ROS production in cells [44]. Mechanistically, HIF-1α activates pyruvate dehydrogenase kinase 1 (PDK1), leading to phosphorylation and inhibition of PDH [102]. Therefore, the role of DJ-1 in regulating metabolic flux through PDH may change depending on the status of HIF-1α activation. Under normoxia or low oxidative stress, a direct interaction between DJ-1 and PDH or mitochondrial localization may upregulate mitochondrial ATP production, while under hypoxia or increased oxidative stress, DJ-1 may block flux through PDH by stabilizing HIF-1α and activating PDK1.

## 3. DJ-1 and Pathophysiological Implications of Altered Metabolism in PD

The importance of metabolism in the pathogenesis of neurodegenerative diseases such as PD is reflected by an increasing number of studies discussing the effect of the nutrition of PD patients [103,104] and that metabolic syndrome can contribute to the pathophysiology of PD [105]. Berry and colleagues found for example that large neutral amino acid levels in the plasma of PD patients were more stable and that the motor performance was superior for patients who had a balanced (5:1) carbohydrate:protein diet compared to patients with unbalanced diets [103], indicating a general role of nutrition and the metabolism in PD pathogenesis. Regarding the involvement of DJ-1 in the regulation of cellular metabolism in the context of PD, there is only little known so far.

In the following, we will present different hypotheses for such a role on the basis of observations made in DJ-1 deficient cells, animal models and in clinical studies. 

GO and MGO are byproducts of lipid peroxidation and glycolysis [106] that, if not quenched, lead to cellular damage via protein and DNA glycation. The two main enzymes responsible for the detoxification of GO and MGO are glyoxalase I and II [106]. DJ-1 as glyoxalase III may contribute to this process [13], especially under conditions where GSH availability is limited (glyoxalase III, as opposed to the glyoxalase I/II system, is GSH-independent) (Section 2.3). A novel deglycase activity has more recently been proposed for DJ-1 [39,107] that repairs glycation damage induced on proteins and DNA by glyoxal and methylglyoxal. Loss of DJ-1 was shown to increase the levels of glycated DNA and DNA strand breaks [38]. However, the proposed deglycase activity of DJ-1 is not yet commonly accepted, as conflicting observations have been reported [108,109]. If confirmed, it would imply, however, that lack of functional DJ-1 could lead to reduced protection from glycation, increased DNA and protein damage and, hence, premature cellular aging via accumulation of advanced glycation end products (AGEs) [38]. A recent clinical study has shown that PD patients have higher plasma levels of carboxymethyllysine, one of the AGEs, compared to healthy controls, suggesting that the roles of AGEs and deglycase function of DJ-1 in PD pathogenesis need to be further studied [110].

A more direct involvement of DJ-1 in the regulation of metabolism was defined by its physical interaction with PDH (Section 2.7), as already mentioned [95]. The consequence of decreased PDH activity in DJ-1-deficient cells is a decreased conversion of pyruvate into acetyl-CoA, which is the main gateway to fuel the TCA cycle [95]. Neurons are metabolically very demanding cells as they need large amounts of ATP to meet their functional requirements, i.e., maintenance of synaptic transmission. Neurons rely on energy production via the TCA cycle and OXPHOS and are incapable of relying solely on glycolysis [111]. Therefore, TCA-OXPHOS impairment due to decreased PDH activity would predictably lead to insufficient ATP production and force neurons to increase their glycolytic flux. Eventually, this could contribute to neuronal cell death. Clinical trials involving a high-energy substrate for ATP, creatine, which bypasses TCA and OXPHOS, have shown that creatine does not improve the clinical outcome of patients with PD [112]. Although it is possible that a sufficient amount of creatine did not reach the brain of the patients, other strategies for restoring energy flux may need to be discovered. In addition, a compensatory increase in glycolytic flux under DJ-1 deficiency could accelerate accumulation of methylglyoxal. Therefore, the effect of DJ-1 on PDH activity could contribute to PD pathogenesis also via AGEs.

DJ-1 also interacts with GAPDH (Section 2.7), although the biological significance and functional consequences of the interaction are unknown. However, the product of GAPDH (1,3-bisphosphoglycerate) is a precursor of 3-phosphoglycerate, which is needed for serine de novo synthesis. As loss of DJ-1 results in decreased serine biosynthesis [88] (Section 2.6), the absence of DJ-1 GAPDH complex in DJ-1-deficient cells could result in decreased 1,3-bisphosphoglycerate synthesis and eventually less 3-phosphoglycerate required for the serine de novo synthesis. Impaired serine biosynthesis was found in PD patient-derived cells [113]. In addition, levels of D-serine in the cerebrospinal fluid of PD patients were lower compared to healthy controls, suggesting that serine metabolism is also important for PD pathogenesis [114].

Interestingly, Gelfin and colleagues found that the treatment of PD patients with D-serine could alleviate behavioral and motor symptoms of the patients [115]. In more detail, D-serine was found to regulate N-methyl-D-aspartate subtype of glutamate receptor (NMDAR) mediated neurotransmission, resulting in an improvement in extrapyramidal symptoms and abnormal involuntary movements, further supporting the importance of serine metabolism in the pathogenesis of PD [115].

In addition to an impairment of serine biosynthesis, glutamine flux was shown to be reduced in the absence of DJ-1, and as glutamine is essential for glutathione synthesis, DJ-1 deficiency impairs the latter on two different levels resulting in an increased sensitivity to oxidative stress [88]. One regulator of oxidative stress that was mentioned earlier to be activated under certain conditions by DJ-1 is the transcription factor Nrf2. In addition to its role in oxidative stress, Nrf2 is increasingly discussed to play a role in mitochondrial bioenergetics and the regulation of expression of metabolic enzymes [116]. Esteras and colleagues discussed pharmacological activation of Nrf2 aiming to restore mitochondrial and metabolic function for the treatment of PD [116]. Nrf2 activation via, for example, Keap1-targeting compounds leads to an increase in substrates that can be used by the TCA cycle and enhances the mitochondrial membrane potential and ATP production [116]. These effects of Nrf2 activation could increase neuronal viability due to their high energy demand in combination with their low glycolytic capacity, suggesting that Nrf2-activating drugs could be of relevance for the treatment of PD [116].

Reduced scavenging of ROS via GSH can have a direct effect on the TCA cycle and OXPHOS as both are regulated by ROS [117]. ROS were shown to stimulate glucose uptake [118]. Under a normal range, ROS stimulate glucose uptake with a beneficial effect on metabolism [117]. However, when ROS levels increase chronically, a vicious cycle of ROS-stimulated glucose uptake and glucose-stimulated ROS production via increased TCA-OXPHOS can be triggered. To counteract this cycle, ROS levels need to be decreased via the DJ-1-mediated pathways or via a decrease in glucose uptake, which will decrease carbon flux into the TCA cycle and depolarize mitochondrial membrane potential if the ATP demand remains high [119]. In addition, chronically high ROS levels in DJ-1 deficiency can eventually lead to mitochondrial dysfunction and apoptosis. In line with these hypotheses, a recent study has shown that DJ-1 deficiency results in decreased mitochondrial membrane potential and decreased ATP production [45]. However, as a decrease in OXPHOS can also be caused by altered TCA flux, these studies are insufficient to capture the entire metabolic status of DJ-1 deficiency.

Clinically, high levels of urate, an antioxidant, in the serum and cerebrospinal fluid of PD patients is associated with slower progression of PD [120,121]. However, randomized clinical trials involving antioxidant vitamin E (tocopherol) have not shown improvement of PD symptoms [122], although more prolonged treatment, once levodopa treatment starts, appeared to slow motor decline [123]. In addition, clinical trials involving mitochondria-targeted antioxidant MitoQ [124] and antioxidant and OXPHOS cofactor CoQ10 [125] have not shown clinical benefit, suggesting that oxidative stress alone may not fully explain the PD pathogenesis.

On a final note, it could be that DJ-1 affects neuronal metabolism in part indirectly via astrocytes. Astrocytes are the major route for brain glucose uptake during periods of strong synaptic activity, indicating that astrocytic glucose uptake is of key importance to neurons [119]. Astrocytes also provide neurons with lactate and glutamine, which are converted into pyruvate and glutamate, respectively, and used in the TCA cycle [118,119]. The loss of DJ-1 in astrocytes could influence neuronal metabolism and viability.

As an outlook, future studies need to define the molecular underpinnings of the role of DJ-1 deficiency in neurodegeneration in PD, and they should include astrocytic, neuronal and co-culture models for metabolic investigations. In addition, there are no studies that investigate the TCA cycle flux in DJ-1-deficient PD cellular or animal models, and the use of experimental tools such as metabolic flux analysis will also shed more light on metabolic effects of DJ-1. Lastly, there is a lack of studies using PD-patient derived cellular models for metabolic studies in the context of DJ-1 deficiency. All of these new tools, when applied to DJ-1 biology, will further clarify the role of DJ-1 in cellular metabolism and their implications in PD pathogenesis.

## Figures and Tables

**Figure 1 cells-10-00347-f001:**
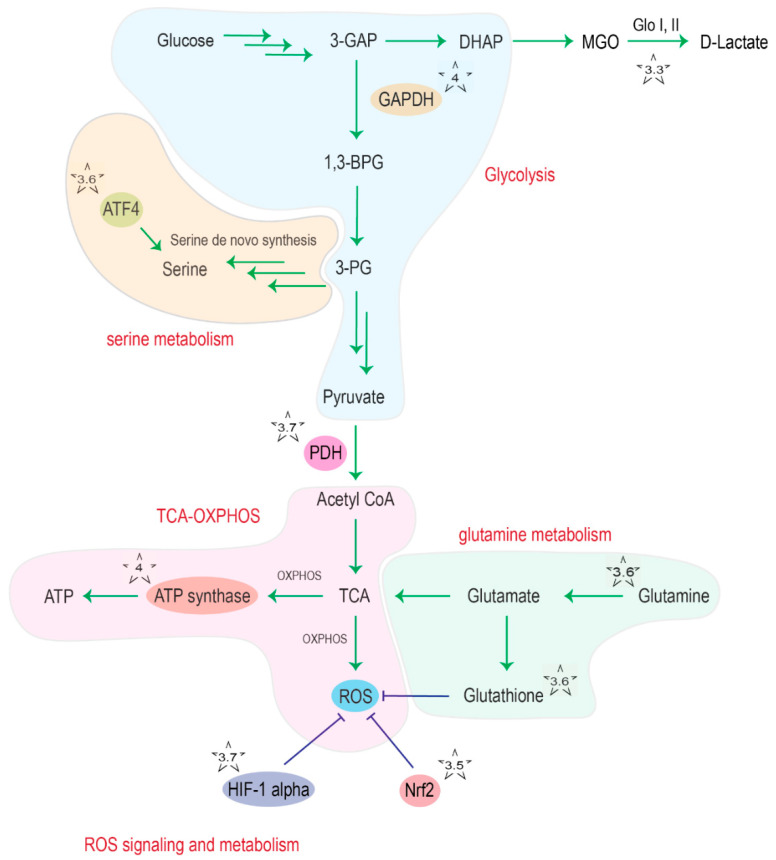
Overview of DJ-1 interaction points (indicated with a star) with cellular metabolism. The chapter discussing the indicated interaction points is given as number in the respective star. Abbreviations: 3-GAP: glyceraldehyde 3-phosphate; DHAP: dihydroxyacetone phosphate; MGO: methyglyoxal; Glo: glyoxalase, GAPDH; glyceraldehyde 3-phosphate dehydrogenase; 1,3-BPG: 1,3-bisphosphoglycerate; 3-PG: 3-phosphoglycerate; ATF4: activating transcription factor 4; PDH: pyruvate dehydrogenase; TCA: tricarboxylic acid cycle; OXPHOS: oxidative phosphorylation; ROS: reactive oxygen species; Nrf2: nuclear factor erythroid 2-related factor 2.

**Table 1 cells-10-00347-t001:** Overview of metabolic alterations depending on DJ-1 status.

Metabolic Alteration	DJ-1 Status or Mutation	Model	Reference
Nucleotide/DNA/RNA glycation	siRNA knockdown	HeLa cells	[38]
Amino acid/protein glycation	C106S, C53S, and C46S DJ-1 mutants	-	[39]
Dicarbonyl-adduct damage	L10P, M26I, A104T, D149A, and L166P	-	[40]
Abnormal mitochondrial morphology mitochondrial/neuronal dysfunction mitochondrial/neuronal dysfunction	Loss of protein	M17 human neuroblastoma cells Mouse embryonic fibroblasts (MEFs) PD patient iPSC-derived neurons	[37] [41] [42] [43] [33]
Compromised mitochondrial uncoupling	Loss of protein	primary murine neurons	[44]
Increased ROS levels	Loss of protein	primary mouse embryonic fibroblasts brains from DJ- KO mice	[45]
Decreased PDH protein levels in DJ-1 KO compared to WT mice	Loss of protein	brain tissue from DJ-1 deficient mice	[46]
Decreased HIF1α level upon hypoxia	Loss of protein	primary cortical neurons derived from DJ-1 KO mouse embryos	[47]
Reduced serine biosynthesis	Loss of protein	LUHMES cells, a dopaminergic neuronal culture model	[48]
Decreased ATF4 transcript expression	Loss of protein	mouse embryonic fibroblasts	[49]

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
