# Peer review of "The Role of DJ-1 in Cellular Metabolism and Pathophysiological Implications for Parkinson’s Disease"

_cells, 2021, doi:10.3390/cells10020347_

Round 1
Reviewer 1 Report
This is a timely article that reviews the pathophysiological roles of DJ-1 in Parkinson's disease. However, it is rather introductory and relies heavily on references to other's findings without much insights. The reviewer believes that the manuscript would benefit from highlighting recent translational studies in greater depth, with particular emphasis in section 4 to be refined into a more coherent piece of work. This will better reflect the current stage of our knowledge in DJ-1 dysfunction and how it relates to PD pathogenesis.
In addition, while broad and attempting to be comprehensive, this review suffers a bit from some pedestrian sections that do not add any real values to the manuscript (e.g. lines 514-521). Several important new findings on how DJ-1 loss-of-function participates in PD-related pathology are also overlooked and shall be discussed (e.g. Liu et al. PNAS 2019; Kumar et al. Commun Biol 2019; Solti et al. Neurobiol Dis 2020). Finally, there are lots of typos and formatting issues (e.g. lnes 33, 49, 88, 98, 220, 225, 412, 442 etc), several language/grammatical mistakes, and the use of acronym in the text is very inconsistent. Please carefully check the text in the revised submission.
Author Response
January 27, 2021
Dear Reviewer,
We would like to thank the reviewers for thorough review of our manuscript and for the helpful and constructive suggestions. We were able to improve on our writing, clarify ambiguities, and focus our review on novel insights. We have summarized our point-by-point response to the reviewers’ comments as well as the changes made in our original manuscript.
Response to the reviewer 1’s comment:
Comment 1: The reviewer believes that the manuscript would benefit from highlighting recent translational studies in greater depth, with particular emphasis in section 4 to be refined into a more coherent piece of work. This will better reflect the current stage of our knowledge in DJ-1 dysfunction and how it relates to PD pathogenesis.
We have now added observations from translational studies which support or negate some of the metabolic functions of DJ-1 described in our review. We included these following clinical studies:
- Diet and PD: lines 503-508
- Deglycase function (levels of advanced glycation end product in PD patients): lines 526-529
- Energy metabolism (clinical trial with creatine supplement): lines 540-547
- Serine metabolism (levels of serine in PD patients and clinical trial with serine supplement): lines 560-575
- Nrf2 as drug target in PD: lines 579-589
- Antioxidant function (levels of antioxidant urate in PD patients and clinical trial with antioxidants vitamin E, CoQ10, and MitoQ): lines 604-611
Comment 2: This review suffers a bit from some pedestrian sections that do not add any real values to the manuscript (e.g. lines 514-521).
We agree with the reviewer 1 that some of the text passages do not add value to the manuscript. We have now rephrased or eliminated pedestrian sentences and expanded upon sentences that require further explanation.
Comment 3: Several important new findings on how DJ-1 loss-of-function participates in PD-related pathology are also overlooked and shall be discussed (e.g. Liu et al. PNAS 2019; Kumar et al. Commun Biol 2019; Solti et al. Neurobiol Dis 2020).
We apologize for omitting these important references. We now cite and discuss recent findings on DJ-1 including Kumar et al. 2019 (section 3.2, lines 166-169), Solti et al. 2020 (section 3.2, lines 173-179), and Liu et al. 2020 (section 3.4, lines 286-294).
Comment 4: Finally, there are lots of typos and formatting issues (e.g. lnes 33, 49, 88, 98, 220, 225, 412, 442 etc), several language/grammatical mistakes, and the use of acronym in the text is very inconsistent.
We have corrected typographical and formatting errors throughout the text.
Having addressed the referee’s comments, we hope you will find our manuscript as meeting the requirements for publication in Cells.
Sincerely,
Pauline Mencke

Reviewer 2 Report
Review
“ The role of DJ-1 in cellular metabolism and pathophysiological implications for Parkinson ́s disease”
The review “The role of DJ-1 in cellular metabolism and pathophysiological implications for Parkinson’s disease” is a well written article summarizing the current literature on the role of DJ-1 on cell metabolism in the context of Parkinson’s disease pathogenesis. The review recapitulates comprehensively several aspects of DJ-1 function based on studies in cellular, non-neuronal and neuronal models and animal models mainly based on a loss-of-function of DJ-1.
My comments are intended to ameliorate the manuscript which was overall a real pleasure to read. Figure 1 is very helpful to provide an overview on the different functions of DJ-1 on cell metabolism and Table 1 helps to understand the variety of models used to investigate DJ-1 functions.
Major comments:
1) Content
A) The last paragraph (4) which is meant to develop some hypotheses on the role of DJ-1 in the context of PD pathogenesis is somewhat repetitive with previous paragraphs and would benefit from a more concise style and more clearly formulated hypotheses. One way to do so might be to better structure the text into paragraphs dealing with one notion in particular, i.e. line 455 -> do not add a paragraph between glycolysis. and Therefore…..
Other examples:
Line 483: what's is the point you want to make? Why is the paragraph cut in two here?
Some text might be better suited for adding into one of the previous sections of the review (3), i.e. all “new” information should be discussed in (3) and comprised in (4) into a discussion without too much repetition. To me, this would favor the pleasure of reading importantly.
B) Line 98: 25% of PD patients with DJ-1 mutations show psychiatric symptoms, cognitive decline, and anxiety compared to patients affected by Parkin or PINK1 mutations34. Is it correct to state this? Please look at this publication: “While patients with Parkin mutations rarely have less cognitive decline and psychiatric symptoms, this seems not applicable to Pink1 mutation carriers” (Cognitive and psychiatric symptoms in genetically determined Parkinson's disease: a systematic review. 27, 229–234 (2020)).
C) It might be of interest to add, i.e. in the introduction, a description of what is known about the expression patterns of DJ-1 in the human and mouse brain – neurons/ astrocytes/ microglia ? Higher expression in certain brain areas compared to others?
D) Line 295 : "Nrf2 can be activated in primary cortical neuronal cultures from DJ-1 deficient mice suggesting that the interaction between Nrf2 and DJ-1 may be cell type specific88". Can you please explain why this suggests cell-type specificity?
E) I wonder if this paragraph is of interest as you do refer to reviews covering other diseases in the introduction. What do we learn from this paragraph that could provide background for the following parts of the manuscript?
-> Line 128: Regarding the involvement of DJ-1 in T2DM, DJ-1-deficient mice were shown to 128 develop impaired glucose tolerance. This indicates that DJ-1 could play a key role in the 129 regulation of glucose homeostasis and that dysregulation of DJ-1 could contribute to 130 T2DM pathophysiology16. In pancreatic beta cells, DJ-1 was shown to protect against 131 oxidative stress and thereby to maintain beta cell viability and insulin secretion16,41–43, 132 suggesting that DJ-1 could play a role in the development of type 1 and 2 diabetes16. Shi 133 and colleagues reported that in skeletal muscle, DJ-1 is involved in the control of energy 134 metabolism44, whereas in adipose tissue, it modulates adipogenesis and obesity-induced 135 inflammation44–47. 136
F) Line 140 “depending on the cell type” -> do you really do this in (4)? Or do you rather discuss the role of DJ-1 “in different cellular models”?
2) Figure
Please recheck the annotations of the paragraphs in the figure
-> Nrf2 and DJ-1 -> 3.5 and not 3.6
Minor comments:
1) Language/Layout
Line 30: alpha/beta
Line 31: add how many alpha-helices here for the understanding later
Line 33/34: oxidative stress is cut into two paragraphs
Line 40: and…first
Line 59: you say this here but then you do add a paragraph on DJ-1 in other diseases later
Line 70: redundant?
Line 80: why this one in particular? It sounds to me as if the others don’t? Maybe you could say: This mutation, for instance, affects DJ-1 protein structure and function…
Line 86: P158? mutation
Line 88: REF
Line 99: add “PD symptoms”, early-onset is not typical
Line 219: point missing
Line 225: like ….mitochondrial
Line 230: is this all Young and colleagues?
Line 233: how you see that mitochondria are inactive on EM?
Line 241: mouse embryonic fibroblast (MEF)
Line 246: reducd
Line 412 : (REF).
Line 441-442 : Additionally, Here the..
Line 509: a recent study have has shown
Line 518: used by the TCA cycle
3) Out of interest:
A) In relation to line 445-8 -> Did anyone look at DNA damage other than oxidation? BER, NER, DSBs ??
B) In relation to line 226 onwards:
In Drosophila, the impact of DJ-1 mutations on mitochondrial quality control seems to involve two other important effectors well known in the context of PD: Pink1 and Parkin. Yang and colleagues demonstrated that Pink1 and Parkin are both implicated in a common pathway that regulates mitochondrial dynamics and cell survival68–70. Downregulation of Pink1 has deleterious effects on the fly phenotype: the flight ability is completely compromised by the flight muscle degeneration and in the brain, the dopamine levels decrease with age. Electron microscopy analysis of tissues revealed mitochondria which are swollen and inactive, also supported by low ATP levels71. The overexpression of Parkin but not DJ-1 rescued loss-of-Pink1 related phenotypes in muscles and brain, further supporting that Parkin acts downstream of Pink172.
In what way then does DJ-1 interact with Parkin and Pink1 concerning mitochondrial homeostasis? DJ-1 apparently did not rescue the LOF of Pink1….
C) Line 302: leading to the protection from oxidative stress induced cellular damage6,90. -> Is it exclusively mitochondrially localized DJ-1 that can be neuroprotective?
Author Response
January 27, 2021
Dear Reviewer,
We would like to thank the reviewers for thorough review of our manuscript and for the helpful and constructive suggestions. We were able to improve on our writing, clarify ambiguities, and focus our review on novel insights. We have summarized our point-by-point response to the reviewers’ comments as well as the changes made in our original manuscript.
Response to the reviewer 2’s comments:
Comment 1 (“1A”): The last paragraph (4) which is meant to develop some hypotheses on the role of DJ-1 in the context of PD pathogenesis is somewhat repetitive with previous paragraphs and would benefit from a more concise style and more clearly formulated hypotheses.
One way to do so might be to better structure the text into paragraphs dealing with one notion in particular, i.e. line 455 -> do not add a paragraph between glycolysis.
We revised section 4 so that we do not simply repeat points raised in the previous sections.
Comment 2 (“1A”): Line 483: what's is the point you want to make? Why is the paragraph cut in two here? Some text might be better suited for adding into one of the previous sections of the review (3), i.e. all “new” information should be discussed in (3) and comprised in (4) into a discussion without too much repetition. To me, this would favor the pleasure of reading importantly.
We have now moved “new information” into section 3 and focused on discussion in section 4.
Comment 3 (“1B”): Line 98: 25% of PD patients with DJ-1 mutations show psychiatric symptoms, cognitive decline, and anxiety compared to patients affected by Parkin or PINK1 mutations34. Is it correct to state this? Please look at this publication: “While patients with Parkin mutations rarely have less cognitive decline and psychiatric symptoms, this seems not applicable to Pink1 mutation carriers” (Cognitive and psychiatric symptoms in genetically determined Parkinson's disease: a systematic review. 27, 229–234 (2020)).
We agree with the reviewer 2 that our statement above does not reflect the current knowledge of psychiatric symptoms in PD patients. We have therefore removed the above sentence.
Comment 4 (“1C”): It might be of interest to add, i.e. in the introduction, a description of what is known about the expression patterns of DJ-1 in the human and mouse brain – neurons/ astrocytes/microglia? Higher expression in certain brain areas compared to others?
We have now added more detailed information concerning the expression pattern of DJ-1 across different cell types in the brain (lines 29-32).
Comment 5 (“1D”): Line 295 : "Nrf2 can be activated in primary cortical neuronal cultures from DJ-1 deficient mice suggesting that the interaction between Nrf2 and DJ-1 may be cell type specific88". Can you please explain why this suggests cell-type specificity?
We agree with the reviewer 2 that the study in primary cortical neurons simply contradicts the observations made by Clements et al. and these two studies alone are insufficient to claim that the interaction between Nrf2 and DJ-1 are “cell type specific.” One would need to examine the interaction between Nrf2 and DJ-1 across multiple cell types to claim “cell specificity.” Therefore, we now simply state that DJ-1 is not required for Nrf2 activation at least in primary cortical neurons (lines 338-340).
Comment 6 (“1E”): I wonder if this paragraph is of interest as you do refer to reviews covering other diseases in the introduction. What do we learn from this paragraph that could provide background for the following parts of the manuscript?
-> Line 128: Regarding the involvement of DJ-1 in T2DM, DJ-1-deficient mice were shown to 128 develop impaired glucose tolerance. This indicates that DJ-1 could play a key role in the 129 regulation of glucose homeostasis and that dysregulation of DJ-1 could contribute to 130 T2DM pathophysiology16. In pancreatic beta cells, DJ-1 was shown to protect against 131 oxidative stress and thereby to maintain beta cell viability and insulin secretion16,41–43, 132 suggesting that DJ-1 could play a role in the development of type 1 and 2 diabetes16. Shi 133 and colleagues reported that in skeletal muscle, DJ-1 is involved in the control of energy 134 metabolism44, whereas in adipose tissue, it modulates adipogenesis and obesity-induced 135 inflammation44–47. 136
We agree that this paragraph does not provide background for the paragraphs that follow. We removed this paragraph as diabetes is already discussed in the introduction.
Comment 7 (“1F”): Line 140 “depending on the cell type” -> do you really do this in (4)? Or do you rather discuss the role of DJ-1 “in different cellular models”?
We agree that we do not have enough information to draw conclusions about cell types. Therefore, we removed the phrase “depending on the cell type.”
Comment 8 (“2”): Please recheck the annotations of the paragraphs in the figure
-> Nrf2 and DJ-1 -> 3.5 and not 3.6
We have corrected the annotations as suggested.
Comment 9 (“Minor comments: Language/Layout”):
Line 30: alpha/beta
Line 31: add how many alpha-helices here for the understanding later
Line 33/34: oxidative stress is cut into two paragraphs
Line 40: and…first
Line 59: you say this here but then you do add a paragraph on DJ-1 in other diseases later
Line 70: redundant?
Line 80: why this one in particular? It sounds to me as if the others don’t? Maybe you could say: This mutation, for instance, affects DJ-1 protein structure and function…
Line 86: P158? mutation
Line 88: REF
Line 99: add “PD symptoms”, early-onset is not typical
Line 219: point missing
Line 225: like ….mitochondrial
Line 230: is this all Young and colleagues?
Line 233: how you see that mitochondria are inactive on EM?
Line 241: mouse embryonic fibroblast (MEF)
Line 246: reduced
Line 412 : (REF).
Line 441-442: Additionally, Here the…
Line 509: a recent study has has shown
Line 518: used by the TCA cycle
We have made changes to the language/layout errors highlighted by reviewer 2.
Comment 10 (“3A”): In relation to line 445-8 -> Did anyone look at DNA damage other than oxidation? BER, NER, DSBs ??
We apologize for the confusion. The DNA damage caused by loss of DJ-1 (as described in Richarme et al. 2017) is DNA glycation and not DNA oxidation. In HeLa cells, DJ-1 knockdown resulted in 3-fold increase in DNA glycation and 12-fold increase in TUNEL-positive cells, which detects double-stranded DNA breaks (DSB). DJ-1 is thought to remove glycated DNA directly although nucleotide excision repair (NER) and mismatch repair (MMR) may also be able to repair glycated DNA. We modified our sentences to clarify this point (lines 520-521).
Comment 11 (“3B”): In relation to line 226 onwards:
In Drosophila, the impact of DJ-1 mutations on mitochondrial quality control seems to involve two other important effectors well known in the context of PD: Pink1 and Parkin. Yang and colleagues demonstrated that Pink1 and Parkin are both implicated in a common pathway that regulates mitochondrial dynamics and cell survival68–70. Downregulation of Pink1 has deleterious effects on the fly phenotype: the flight ability is completely compromised by the flight muscle degeneration and in the brain, the dopamine levels decrease with age. Electron microscopy analysis of tissues revealed mitochondria which are swollen and inactive, also supported by low ATP levels71. The overexpression of Parkin but not DJ-1 rescued loss-of-Pink1 related phenotypes in muscles and brain, further supporting that Parkin acts downstream of Pink172.
In what way then does DJ-1 interact with Parkin and Pink1 concerning mitochondrial homeostasis? DJ-1 apparently did not rescue the LOF of Pink1….
We agree with the apparent contradiction. In one study, DJ-1 overexpression rescued Pink1 mutant phenotype while in another study, it did not. The level of overexpression appears to be one of the important factors for the success of rescue. Moreover, DJ-1 can directly activate Parkin through S-nitrosylation. We have modified our sentences to reflect this point (lines 252-261).
Comment 12 (“3C”): Line 302: leading to the protection from oxidative stress induced cellular damage6,90. -> Is it exclusively mitochondrially localized DJ-1 that can be neuroprotective?
Based on Junn et al. 2009, expression of mitochondrially localized DJ-1 have “enhanced” neuroprotection (55% protection based on LDH release) while cytosolic and nuclear localized DJ-1 still have neuroprotection (30% protection). Therefore, DJ-1 in all three compartments are neuroprotective with mitochondrial DJ-1 having slightly higher neuroprotection. We modified our text accordingly to clarify this point (lines 347-350).
Having addressed the referee’s comments, we hope you will find our manuscript as meeting the requirements for publication in Cells.
Sincerely,
Pauline Mencke
